# NEURAL ELECTROSTATICS:
# A 3D PHYSICS-INFORMED BOUNDARY ELEMENT POISSON EQUATION SOLVER

## ABSTRACT

Electrostatics solvers relate an imposed voltage to a corresponding charge density. Current classical methods require fine discretization and scale poorly due to the construction of a large linear system of equations. We recast the problem using neural networks and introduce neural electrostatics, a hybrid 3D boundary element method (BEM). By using the boundary element form, we are able to overcome many shortcomings of previous neural solvers, such as learning trivial solutions and balancing loss terms between the domain and boundary, at the cost of introducing a large integral containing a singular kernel. We handle this singularity by locally transforming the integral into polar coordinates and applying a numerical quadrature. We also show that previous neural solver sampling methods are unable to minimize the physics-informed residual, and propose a variational adaptive sampling method. This technique is able to reduce mean absolute error by 5 times, while keeping training time constant. Extensive scaling and ablation studies are performed to justify our method. Results show that our method learns a charge distribution within 1.2 $pC/m^2$ of mean absolute error from a classical BEM solver, while using 25 times fewer rectangular elements.

## 1 INTRODUCTION

The computational modeling of electrostatics, relating electric fields to charge density, is critical in the design and application of a wide range of modern technologies. Simulation of capacitor and battery design (Zhan et al., 2017) relates the storage and release of electrical energy to material and environmental properties, leading to improved storage solutions. Capacitive touch sensors, found in smartphones and tablets, rely on simulations to adjust sensitivity and responsiveness reaching to changes in electric fields (Zhou et al., 2020). As we shrink mechanical designs and electrical circuits, the electromagnetic forces dominate and their simulation is required for micro-electro-mechanical systems compatibility (Muhkopadhyay & Majumdar, 2006) and static electric discharge protection (Voldman & Gross, 1994). The advancement of computational tools for electrostatic field modeling is a requirement for future technologies, allowing for greater accuracy, efficiency, and innovation.

The last 80 years has seen the proliferation of computational electrostatics methods, from explicit methods like the finite difference method (FDM) (Burden et al., 2016) to implicit methods like the finite element method (FEM) (Burden et al., 2016) and the boundary element method (BEM) (Brenner & Scott, 2003). BEM is typically referred to as the method of moments (MoM) (Harrington, 1993; Gibson, 2014) in the electrical engineering literature. Each method has its strengths and weaknesses, but they all fail to extend to large computational domains because of discretization and memory requirements. Even BEM, which reduces the dimensionality of the problem, struggles to scale to domains of practical interest, such as rooms or large vehicles. Techniques such as the multi-level fast multipole method (Song, 1997) or the adaptive cross approximation Zhao et al. (2005) can be applied to aid in scaling, but are still limited by the discretization and roots in classical techniques. We seek to address this scaling problem instead by reformulating the method of moments as a function regression problem using neural networks and stochastic gradient descent (Kiefer & Wolfowitz, 1952).

The growing field of scientific machine learning combines the advances of modern machine learning with classical techniques for solving partial differential equations (PDE). The straightforward application of these principles is the physics-informed neural network (PINN) (Raissi et al., 2017; Cuomo et al., 2022; Wang et al., 2023). PINNs leverage the automatic differentiation of neural networks to apply differential operators on the network and build a loss between the mapped network and the forcing function at sampled points in the domain and boundary. The PINN method, however, often learns trivial solutions, is slow, and does not scale outside its original training domain. Many variations of the original PINN method have been proposed (Lu et al., 2019; Ramabathiran & Ramachandran, 2021), but meaningful extensions are difficult. A promising approach is variational PINNs (Kharazmi et al., 2021; 2020) and the combination of FEM theory with deep learning (Gao et al., 2022; Yu, 2017; Aylwin et al., 2023). By introducing a test function set, the PDE is recast into a weak form, trading additional network evaluations for reduced regularity requirements on the network. The variational form even reduces memory, as only lower order derivatives are necessary. These applications, however, are normally evaluated in 1D or 2D domains, and do not immediately remove the possibility of trivial solutions. By using a boundary element method and recasting the problem with the method of moments, we are able to solve the electrostatics equation on arbitrary 3D geometry and reconstruct the field anywhere in space. By combining the classical methods with deep learning, we present a flexible and easy to evaluate basis function in the form of a feed forward neural network. Specifically, we make the following *contributions*:

- We present a novel 3D boundary element solver that uses a neural network basis function to represent the charge density along a boundary. By formulating the problem in this manner, we avoid learning trivial zero solutions to the PDE and only need to minimize a single loss function, as there is no separation between domain and boundary. Once solved, the charge density can be used to compute the electric field anywhere in an infinite domain.

- We develop and empirical demonstrate a singularity removal technique. By locally transforming the integral into polar coordinates, we robustly handle the $\frac{1}{r}$ term that arises in the boundary element form of the Poisson equation. Though this singularity is well explored in classical methods, we show that similar methods can be applied to neural solvers without hindering backpropagation.

- We introduce a variational adaptive sampling technique that decreases error without increasing training time. In our experiments, the mean absolute error, as compared to the finely discretized classical BEM solver, is decreased by 5 times as compared to existing loss functions, while training time is held constant.

- We validate our method on the standard problem of a flat metal plate held at a constant voltage. We perform both scaling and ablation studies to both justify our decisions and show that the number of test functions is decoupled from the size of the network. Our baseline solution is able to represent the charge density within $1.2\,pC/m^2$ of mean absolute error from a reference BEM solver with 25 times the rectangular elements. Training to a reasonable error on this problem takes approximately 1 hour, while inference of the charge density is incredibly fast. This fast inference time is important as we must integrate the network to compute fields at any point in space.

## 2 RELATED WORK

Since the original work of Harrington (1993), the method of moments has been the de facto standard for high fidelity solvers in electromagnetics. Assuming isotropic materials, the method is able to reduce PDEs from 3D volumes to 2D surface integrals. When the domain is large, or in many cases infinite, this method is invaluable. The surface integral equations can then be solved by standard BEM methods (Brenner & Scott, 2003), first by expanding the current or charge density by a set of basis functions and then enforcing boundary conditions through a set of test functions. MoM, in its standard form, produces a dense matrix which grows quadratically with the discretization of the basis functions. Therefore, large objects require large matrices, often producing matrices so large they cannot fit in system memory. Multiple techniques seek to address this scaling issue by discretizing space, recognizing that fields die off with increasing distance. The Multi-level Fast Multipole Method (MLFMM) (Song, 1997) interacts these grids iteratively in a hierarchical structure. These interactions must be derived for the underlying equation. The Adaptive Cross Approximation (ACA)

(Zhao et al., 2005), in contrast, is a purely linear algebra technique that takes advantage of low rank sub-blocks. ACA also requires no iteration and can be solved directly using block lower-upper (LU) Decomposition (Burden et al., 2016). These scaling method are difficult to implement, and though scaling is greatly improved, very large geometries (i.e., room scales) are computationally infeasible. Instead of improving the linear solve, we seek to replace the basis function set with a neural network that can easily adapt to arbitrary geometry.

Reducing the 3D problem to a 2D problem comes at the cost of a newly introduced singularity encountered during integration. If the singularity it not properly handled, training will not converge and the solver will often produce NaNs. Singular kernels have been extensively addressed in the classical literature but are relatively unexplored in the context of neural basis functions. Previous work has simply ignored the singularity (Ruocco et al., 2023), which allows training to converge, but introduces error. In contrast, classical singularity removal techniques separate the integral into far and near terms and include the contribution of the singularity and the surrounding region. Monte Carlo integration is theoretically robust to point singularities: the probability that the singular point is sampled is zero. In practice, samples close to the singularity can produce NaNs, so at the cost add bias, a small exclusion region is introduced around the singularity. Instead of working around the singularity, other method seek to remove it from the integral. The near term can be specially treated (Taylor, 2003; Ylä-Oijala & Taskinen, 2003), while the far term integral can be easily integrated with a numerical quadrature. The simplest method casts the integral into polar coordinates, which introduces a radial term canceling out the singularity. We investigate the Monte Carlo exclusion and coordinate transform techniques applied to a neural network basis and derives bounds for their approximation error.

Engineers have begun tackling electrostatics problems with scientific machine learning, but none have yet to address the boundary element formulation or domain scaling. A straightforward application of machine learning techniques to electrostatics was through FDM, and its electrodynamics parallel the finite difference time domain (FDTD) method (Tang et al., 2017; Guo et al., 2019; Yao & Jiang, 2018). Adjacent work on BEM based PINNs (Lin et al., 2021; Nagy-Huber & Roth, 2024; Sun et al., 2023) only considered 2D domains and a boundary integral derived from potential theory not used in electrostatics problems. Just like their classical counterparts, the machine learning based finite different methods required rigid grids and suffered from stability issues. Practitioners then turned towards variational methods in the same vein as FEM, BEM, and MoM (Key & Notaros, 2020; 2021). These methods mirror the classical methods and build off the more general VPINN (Kharazmi et al., 2019), Deep Ritz (Yu, 2017), and Deep Galerkin methods (Sirignano & Spiliopoulos, 2018), by casting the PDE in a weak formulation. The weak form reduces the regularity requirements on the neural network and makes the solution more stable at the expense of more network evaluations and more difficult convergence. We extend these variational methods to the electrostatics problem on an arbitrary boundary in 3D.

## 3 BACKGROUND AND NOTATION

We briefly highlight the notation (following Goodfellow et al. (2016)) used in the rest of the paper, as well as provide the necessary background in electrostatics and physics-informed neural network for understanding our contributions. We deviate slightly from this notation when discussing the electric field, $\mathbf{E}$. This is a vector not a matrix, but we want to follow the common notation in physics and electrical engineering. $\Omega$ denotes a region, while $\Gamma$ is a boundary. We commonly refer to $\mathbf{r}$ and $\mathbf{r}'$. These are points in Euclidean space. The non-primed coordinate is referred to as a test point, where the field is measure, whereas the primed coordinate is a source point.

### 3.1 ELECTROSTATICS

Electrostatics relates an electric field, $\mathbf{E}$, to a charge density, $\rho_e$, at individual points in space. When the fields are not time varying, the electric and magnetic fields decouple, and the curl of the electric field becomes zero. An irrotational vector field can be further simplified to a scalar function, which we will denote $\phi$ by convention. This scalar function is called the voltage and is related to the electric field by $\mathbf{E} = -\nabla\phi$. Plugging the voltage equation into Gauss's Law, which equates the divergence of the electric field with the scaled charge density, we get the equation of interest in electrostatics,

$$\nabla^2 \phi(\mathbf{r}) = -\frac{\rho_e(\mathbf{r})}{\epsilon_0}. \tag{1}$$

$\epsilon_0$ is a physical constant representing the permittivity of free space. Unlike common applications with the Poisson equation, where $\phi$ is unknown and $\rho_e$ is known, the electrostatics problems flips this, as we can more easily measure voltage than charge density. The differential form of Equation 1 can be converted to an integral form using the Green's function (Appendix B), giving

$$\phi(\mathbf{r}) = \iiint_\Omega G(\mathbf{r}, \mathbf{r}') \frac{\rho_e(\mathbf{r}')}{\epsilon_0} d\mathbf{r}' = \iiint_\Omega \frac{\rho_e(\mathbf{r}')}{4\pi\epsilon_0 |\mathbf{r} - \mathbf{r}'|} d\mathbf{r}'. \tag{2}$$

We focus our work on the exterior problem, where the boundary condition is that the field goes to 0 at infinity. This is trivially satisfied by the fundamental solution of the Laplace operator.

## 3.2 VARIATIONAL PHYSICS-INFORMED NEURAL NETWORK

Assume $L$ is a differential operator, $f$ is the forcing, $\Omega$ is the domain, and $g$ is a Dirichlet condition imposed on the boundary $\Gamma$. PINNs solve equations of the form

$$Lu(\mathbf{r}) = f(\mathbf{r}) \; ; \; \mathbf{r} \in \Omega \; \text{ with } \; u(\mathbf{r}) = g(\mathbf{r}) \; ; \; \mathbf{r} \in \Gamma \tag{3}$$

by modeling $u(\mathbf{r})$ as a neural network, $u_\theta(\mathbf{r})$. The training dataset is then built up from samples $\mathbf{r}_i \in \Omega$. Care needs to be taken to adequately sample both the domain and the boundary (Wang et al., 2023). For Variational Physics-Informed Neural Networks, we recast the problem to its weak, or variational, form by introducing a set of test functions, $\{v_1, \ldots, v_N\}$. Training then swaps random point sampling with integration across the set of test functions. The loss functions are then given by

$$L_\Omega = \frac{1}{N_\Omega} \sum_{i=1}^{N_\Omega} ||\langle v_i, Lu_\psi(\mathbf{r}_i) \rangle - \langle v_i, f_\theta(\mathbf{r}_i) \rangle|| \; \text{ and } \; L_\Gamma = \frac{1}{N_\Gamma} \sum_{j=1}^{N_\Gamma} ||u_\theta(\mathbf{r}_j) - g(\mathbf{r}_j)||. \tag{4}$$

The regularity requirements place on $u_\theta$ in Equation 3 are reduced by the variational form through the application of integration by parts. This reduction improves memory requirements and alleviates differentiability requirements on the activation function.

## 4 NEURAL ELECTROSTATICS

### 4.1 LEARNING PROBLEM FORMULATION

If we cast the electrostatics problem in the original PINN methodology, we would need two neural networks: one for the voltage and one for the charge density. These two networks would then be trained at known boundary points to enforce the PDE. The singularity in Equation 2 would not be present, but needing to train two networks simultaneously increases the complexity of the problem. This problem formulation also imposes additional constraints on the network representing the voltage, $\phi(\mathbf{r})$. First, the activation function used needs to be three times differentiable, twice for the Laplace operator and once more for backpropagation. This means that ReLU activation cannot be used. Training of the voltage network is also computationally expensive, as the third derivative is needed. As such, the memory needed to update the weights grows as the size of the network cubed. This problem formulation also tends towards the trivial case, where both networks learn the zero solution everywhere. Instead, we cast the Poisson equation in its boundary element form and define the variational residual loss (details in Appendix C), setting $v_i$ as a rectangular test function, gives

$$r_i = \langle v_i(\mathbf{r}), \phi(\mathbf{r}) \rangle - \langle v_i(\mathbf{r}), \iint_\Gamma \frac{\rho_e(\mathbf{r}')}{4\pi\epsilon_0 |\mathbf{r} - \mathbf{r}'|} d\mathbf{r}' \rangle. \tag{5}$$

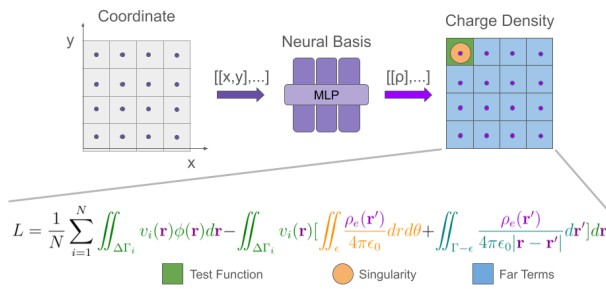

Figure 1: We propose neural electrostatics in this diagram. Samples from across the 3D mesh are passed into the MLP, which predicts the corresponding charge density. Using the variational, rectangular test functions, integrals are approximated using these points. A test function, green, is randomly selected. The singularity removal, orange, is performed in the near term, whereas the far term, blue, is handled with no special treatment. All integrals are performed using either Gauss-Legendre or Newton-Cotes quadratures. Test functions are adaptively drawn to help decrease error where it is highest.

$v_i(\mathbf{r})$ is defined as 1 when $\mathbf{r} \in \Gamma_i$ and 0 otherwise. Different loss functions can be formed from this residual. We use mean squared error loss, giving

$$L = \frac{1}{N} \sum_{i=1}^{N} ||r_i||^2. \tag{6}$$

A key benefit of this approach is that there is no separation between domain and boundary samples, as the entire problem is restricted to the boundary. Therefore, we do not need to consider loss balancing schemes typically found in PINNs.

### 4.2 Coordinate Transform Singularity Removal

In boundary element methods for the Laplace (and the similar Helmholtz operator), the singularity that arises is often a point of difficulty. There have been many techniques developed in the FEM literature (Colton & Kress, 2013); however, their extension to PINN like problems has not been well established. We emphasize that the point singularity is purely a property of the Green's function and is not present in either $\phi$ or $\rho_e$. This fact is important because the network itself does not need to represent this singularity, but rather the training algorithm must be robust to it. We provide the stability by separating the integral into near and far domains and then transforming the near integral into a frame where the singularity is no longer a problem.

The integrals presented so far are in the typical Euclidean frame. By converting to polar coordinates centered at the singularity, the change of variables will introduce a radius term in the numerator. The integral then becomes

$$\iint_\Gamma \frac{\rho_e(\mathbf{r}')}{4\pi\epsilon_0|\mathbf{r}-\mathbf{r}'|} d\mathbf{r}' = \iint_{\Gamma_\circ} \frac{\rho_e(\mathbf{r}')}{4\pi\epsilon_0|\mathbf{r}-\mathbf{r}'|}|\mathbf{r}-\mathbf{r}'| dr d\theta = \iint_{\Gamma_\circ} \frac{\rho_e(\mathbf{r}')}{4\pi\epsilon_0} dr d\theta. \tag{7}$$

The transformed integral appears to be much easier to deal with, but it has a major drawback. Instead of integrating over the surface in a Euclidean coordinate system, we must now integrate over the bounds in the polar coordinate system. These bounds are more difficult to determine for complex geometries, whereas the original form can easily be evaluated from a triangular or quadrilateral mesh. To alleviate this issue, we separate the integral into two domains: one near and one far. We note the near domain with $\epsilon$. We can view the near domain as locally planar and integrate over a small square of side length $2\epsilon$ centered at the singularity. This recasting gives

$$\iint_{\Gamma} \frac{\rho_e(\mathbf{r'})}{4\pi\epsilon_0 |\mathbf{r} - \mathbf{r'}|} d\mathbf{r'} = \iint_{\epsilon} \frac{\rho_e(\mathbf{r'})}{4\pi\epsilon_0 |\mathbf{r} - \mathbf{r'}|} d\mathbf{r'} + \iint_{\Gamma-\epsilon} \frac{\rho_e(\mathbf{r'})}{4\pi\epsilon_0 |\mathbf{r} - \mathbf{r'}|} d\mathbf{r'}$$

$$= \iint_{\epsilon} \frac{\rho_e(\mathbf{r'})}{4\pi\epsilon_0} dr d\theta + \iint_{\Gamma-\epsilon} \frac{\rho_e(\mathbf{r'})}{4\pi\epsilon_0 |\mathbf{r} - \mathbf{r'}|} d\mathbf{r'}. \quad (8)$$

As the neural network has no closed form integral, we evaluate the near domain with a quadrature rule and leave the evaluation of the far integral to the next section. We chose the Gauss-Legendre quadrature Burden et al. (2016) in our work, as it is a common choice in numerical applications. Because we consider a square domain, to integrate in polar coordinates, we must separate the integral again into 4 quadrants. The near integral now takes on a form that can be implemented on a computer:

$$\iint_{\epsilon_\circ} \frac{\rho_e(\mathbf{r'})}{4\pi\epsilon_0} dr d\theta \approx \frac{1}{4\pi\epsilon_0} \sum_{q=1}^{4} \sum_{i=1}^{N_\theta} \sum_{j=1}^{N_r} w_i w_j \rho_e(\mathbf{r'} + \tilde{\alpha}_{ji}[\cos(\tilde{\alpha}_i), \sin(\tilde{\alpha}_i)]) \quad (9)$$

More details on the evaluation of this integral is found in Appendix D.

### 4.3 INTEGRATION

With the singularity mitigated, we now consider how to evaluate the integral still present in the loss term (Equation 5).

**Far Term.** A composite Newton-Cotes quadrature (Burden et al., 2016) is well suited to evaluations on a fixed grid, like a mesh would provide. To expand to large domains, adaptive quadratures can be used. The quadrature order can be decreased as the distance between the source and test points grows. In the present work, we only consider a fixed quadrature of 0 order. This is equivalent to the midpoint rule taught when first learning calculus. We consider the open form of the quadrature, as it does not require evaluation of the function along the boundary. This is important, as sharp discontinuities, such as edges on a flat plate, cannot be evaluated. The far term integral, thus, takes the form

$$\iint_{\Gamma} \frac{\rho_e(\mathbf{r'})}{4\pi\epsilon_0 |\mathbf{r} - \mathbf{r'}|} d\mathbf{r'} = \sum_{i=1}^{N} \iint_{A_i} \frac{\rho_e(\mathbf{r'})}{4\pi\epsilon_0 |\mathbf{r} - \mathbf{r'}|} d\mathbf{r'} \approx \frac{\Delta A}{4\pi\epsilon_0} \sum_{i=1}^{N} \frac{\rho_e(\mathbf{c}_i)}{|\mathbf{r} - \mathbf{r}_i|}. \quad (10)$$

Where $\mathbf{c}_i$ is the center of each rectangle and $\Delta A$ is the corresponding area. As all evaluations of the network during training need this integral, these points can be evaluated a single every epoch, given there is enough memory available. The near terms can then be masked off and their contribution replaced by the coordinate transform.

**Test Function.** With the use of a rectangular test function, another surface integral needs to be performed. With no closed form solution for the inner, singular kernel integral, we perform a quadrature on the outer test function. Due to the simple nature of the rectangular test function, we apply a composite midpoint Newton-Cotes quadrature to approximate. This evaluation gives,

$$\langle v_i(\mathbf{r}), \iint_{\Gamma} \frac{\rho_e(\mathbf{r'})}{4\pi\epsilon_0 |\mathbf{r} - \mathbf{r'}|} d\mathbf{r'} \rangle \approx wh \sum_{i=1}^{N_w} \sum_{j=1}^{N_h} \iint_{\Gamma} \frac{\rho_e(\mathbf{r'})}{4\pi\epsilon_0 |\mathbf{c} + \hat{\mathbf{w}}\alpha_i + \hat{\mathbf{h}}\alpha_j - \mathbf{r'}|} d\mathbf{r'}. \quad (11)$$

$w$ and $h$ are the width and height of each subdivision, while $\mathbf{c}$ is the center of the rectangular basis function. $\hat{\mathbf{w}}$ and $\hat{\mathbf{h}}$ are tangent unit vectors along the width and height, respectively. The quadrature points, $\alpha_{i,j}$ are evenly spaced along each dimension going from $[-\frac{w}{2} + \frac{w}{2N_w}, \frac{w}{2} - \frac{w}{2N_w}]$ and $[-\frac{h}{2} + \frac{h}{2N_h}, \frac{h}{2} - \frac{h}{2N_h}]$ along the width and the height. More sample efficient methods could be used, such as Gauss-Legendre quadrature; however, this was found to be unnecessary in the present application. By using this grid structure, it made calculation of the far term easier, as the singular region radius could be set to $\min(\frac{w}{2N_w}, \frac{h}{2N_h})$.

## 4.4 VARIATIONAL ADAPTIVE SAMPLING

PINN methods can often struggle to minimize the PDE residual loss with mean squared error (MSE) alone. MSE forces the network to obey the PDE on average; however, the PDE requires exactness or else a nearby erroneous solution will be learned. $L_1$ loss can help, but is a more difficult loss function due to discontinuous derivatives at 0. Other works have used a hybrid loss function that combines MSE and $L_\infty$. The MSE term minimizes the residual overall, while the $L_\infty$ applies a soft constraint to outliers. This adds additional complexity because the loss terms need to be balanced. Also, $L_\infty$ only penalizes the largest outlier, reducing efficiency of the approach. Instead, We propose an adaptive sampling scheme that uses MSE while additionally decreasing loss without increasing training time.

Taking inspiration from the residual adaptive resampling (Lu et al., 2019) algorithm, we improve loss by sampling where the error is the highest; however, instead of gradually increasing the data set (and, therefore, increasing the training time), we temporarily remove the test functions with the lowest error (below a hyperparameter $\delta$) and train for a set number of epochs. After which, we reintroduce all the test functions back into the training set. This continual contraction and expansion of the training set allows the network to adapt to the regions with the largest error.

This error calculation requires little additional computation and no extra evaluations of the network, as it leverages the output directly from the training step. We also only reduce the training set every $N_d$ epoch, so the small computation is amortized across the whole training run. By reducing the training dataset, the training time temporarily decreases as fewer evaluations are required until $N_r$ when we add back the removed test functions. This is especially important as each evaluation requires integrating across the entire domain. The full algorithm is highlighted in Appendix G.

## 5 RESULTS

### 5.1 PROBLEM AND METRICS

For evaluation, we consider the problem of a 2-meter by 2-meter flat plate at constant voltage. The plate lies in the XY Plane centered at $[0, 0, 0]$, with the normal facing along the positive z-axis. Though simple, this problem poses a difficult challenge for other PINN formulations, as it requires integration through a singular kernel. We validate our results using a textbook BEM solver (Gibson, 2014) (Figure 2a), both at a fine "ground truth" discretization of 50 by 50 elements and a discretization matching that used in the neural electrostatics algorithm. The comparison at coarser discretization allows us to compare our method against the BEM solver at a set discretization. Both evaluations are calculated as the average $L_1$ distance between the solutions at the center of each rectangular function. We estimate PDE residual error at each test function. We compute the absolute error between the voltage inner product and the charge density inner product. To obey the physical relationship, we want the error between the left and right hand terms of Equation 5 to be as small and low variance as possible. For a good solution, we want to perform well on both metrics, as performing well on only one, could mean we found an incorrect, non-physical nearby solution to the PDE instead. We present additional evaluation in Appendix I.

### 5.2 SCALING STUDY

In this section, we seek to understand how the size of the network and number of rectangular test functions influences our results. 9 network sizes are considered with hidden layer depths of 1, 2, or 3 and hidden layer widths of 32, 64, 128. The number of test functions are also varied from 25 to 100 to 225. We compare the performance of each evaluation against the BEM results shown in Figure 2a. For every evaluation, the test function quadrature order is set to 3 along both x and y. The grid integration scheme matches this sampling and aligns itself with the test functions. The near term singularity then uses a 2 point quadrature along $r$ and $\theta$ in each quadrant.

From Figure 2b, we find that the performance of the method in this problem is largely independent of the size of the network. Therefore, we are able to keep the network small and still accurately represent the solution. Small networks also mean faster evaluations and training. The number of test functions, however, does impact performance. We see a large decrease in error when moving from 25 to 100 test functions. There is then a much smaller decrease in error moving from 100 to 225

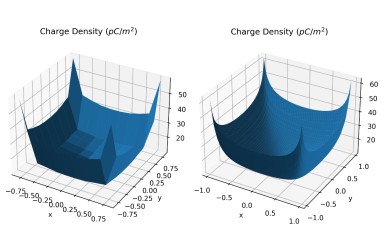 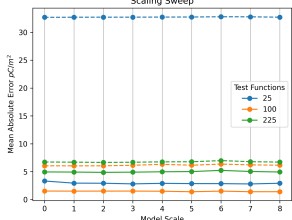

(a) BEM Solutions at 100 and 2500 test functions.    (b) Scaling sweeps across basis network size.

Figure 2: This figure shows both the reference BEM solution used for comparison (a) and the scaling analysis results (b). For scaling, the x-axis represents the increasing size of the network. The colors map to different counts of rectangular test functions. The solid line is the error relative to the truth, while the dotted line is the error relative to the BEM solver at the same discretization. From this plot, we find that the problem is not sensitive to the size of the problem. Rather, the number of test functions dictates the final error. This means that we can keep the network small for faster evaluation.

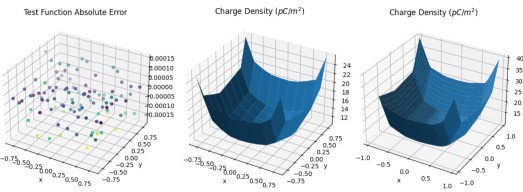

Figure 3: Baseline results computed using neural electrostatics for comparision in ablation studies. Hyperparameter details are found in Table 1. The left shows the distribution of errors across test functions. The middle figure plots the charge density sampled at the same intervals used during sampling. The right is finely sampled to compare with the ground truth solution.

test functions, showing diminishing returns as the number of test functions is increased. Comparing Figures 2a and 3, we see that even with 100 test functions, the charge density is well characterized. We notice a slight increase in error between the 225 and 100 test functions. It is believed that this is because there are more sample points near the boundary discontinuity, which is inherently difficult to characterize in any numerical solver. There is a fundamental decoupling of the basis function and test function sizes in our method that allows one to scale independently of the other, whereas classical techniques require an equal number of test and basis functions to construct a full rank matrix. Future work could leverage this decoupling to solve larger problems with far fewer basis and test elements than is traditionally needed.

## 5.3 ABLATION STUDIES

**Baseline.** The scaling study showed that the error is largely independent of the basis network size. So, we select the median network test with a width of 64 and hidden layer depth of 2. 100 rectangular test functions are used, as 100 had a large improvement over 25 but was only marginally worse than 225. 100 was selected as a representative middle ground. We include other training parameters in Table 1.

**Test Function.** We compare three classes of test functions: collocation, point matching, and the baseline rectangular. Collocation is what is typically used in PINNs, including the previous BEM formulations (Sun et al., 2023; Lin et al., 2021; Nagy-Huber & Roth, 2024). The method enforces the residual to be equal at discrete, random points over the domain. In our test, we use the residual adaptive resampling (RAR) (Lu et al., 2019) method. RAR importance samples the residual function by adding points with the highest error to the training set. Point matching is similar to both the collocation method and the rectangular test function used in our baseline. Instead of testing at random locations, point matching evaluates the residual on a discrete grid.

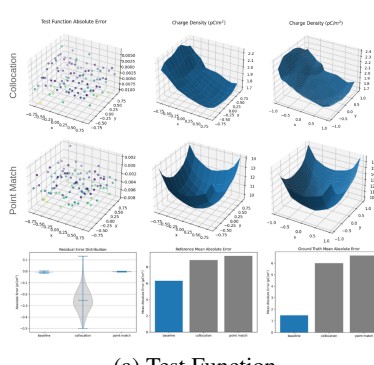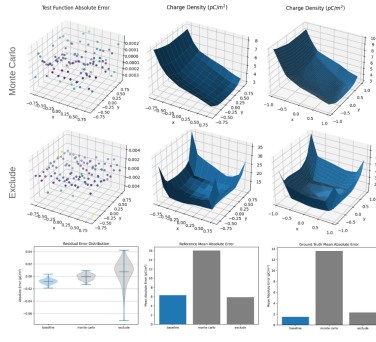

(a) Test Function (b) Integration and Singularity Removal

Figure 4: Ablation studies for test function (a) and the combination of integration and singularity removal (b). The collocation method and Monte Carlo integration techniques are unable to learn the rough shape of the solution and have high mean absolute errors. The collocation method also shows wide variance, so the errors are not uniformly minimized. Point matching performs better than the collocation method, but is unable to capture the large discontinuities around the edges. Point matching then has low variance, but high MAE. Lastly, the exclusion technique is able to capture the rough shape but adds too much bias to the integral. This results in high variance, but low MAE. Our baseline, outperforms all of these techniques with both low variance and low MAE.

Figure 4a shows that the collocation method is unable to learn the correct charge distribution and does not show the expected cup shape. This is corroborated by our metrics. The error distribution for the collocation technique has a mean far from 0 and a wide variance, meaning that the method was not able to optimize the PDE residual. Similarly, we see large reported mean absolute errors as compared to the BEM solver. Compared to our baseline, for a similar number of data points, we find that the variational scheme is able to reduce the PDE error more effectively. We also note that RAR continually adds points to the dataset, which can drastically impact training, as these additional points require a singularity removal and integrating across the boundary. Our variational adaptive sampling method instead reduces the number of evaluations while also decreasing error.

The point matching technique performs better than the collocation method and is able to produce the characteristic cup shape of the charge density. The residual distribution is also close to 0 and has a small variance; however, we see that there is a large mean absolute error with the reference solver. This is because the point matching technique is unable to capture the sharp spikes at the edge of the plate, and the overall range is only 4 picocoulombs per square meter. The rectangular test function in the baseline solver is able to both learn the correct cup shape and capture the sharp rise at the edges. This results in a favorable residual error distribution and lower mean absolute error than the other techniques.

**Singularity Removal and Integration.** We compare our singularity removal technique against a Monte Carlo exclusion method (Appendix F) and combined coordinate transform and Monte Carlo integration method (Appendix E), with cutoff distances of $\epsilon = 0.001$ and $\epsilon = 0.05$, respectively. Figure 4b shows that neither of these methods are able to produce a network that captures the charge density. The disc and Monte Carlo integration technique is the worst performer, as it is unable to even represent the cup shape of the charge density. In our metrics, this technique has a low residual variability and 0 mean, but its mean absolute error is very high. We believe that this is due partly to the random sampling in the technique, as the network needs to characterize that noise as well. Samples close to the edge of the plate could also cause problems, as there is step discontinuity in the voltage.

The exclusion region method produces a qualitatively better solution, but there is obvious deformation in the charge density function. The cup shape is there, but there is no symmetry or expected smoothness. Our metrics reflect this. The technique has a low mean absolute error (even rivaling our baseline), but the residual distribution varies highly and non-uniformly. The network is better able to represent the stochasticity, but the bias of removing the singularity's contribution leads to an incorrect solution.

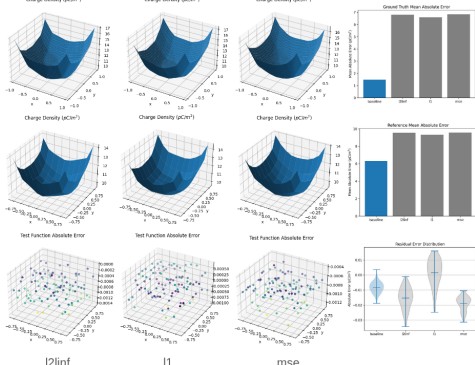

Figure 5: Results for ablation study on different loss functions. We see that all three loss functions produce similar results. The rough shape of the charge distribution is captured, but the large discontinuities on the edge of the plate are missed. The adaptive sampling technique, in the baseline, is able to better capture this discontinuity, resulting in lower error variance and smaller mean absolute error (bottom).

**Loss Function.** Finally, we compare our variational adaptive training scheme to common loss functions found in PINNs. We compare our method against mean squared error (MSE), mean $L_1$ distance, and a combined MSE and $L_\infty$ distance. Details can be found in the past section on the variational adaptive training method. Figure 5 shows that all loss functions are able to qualitatively match the shape of the charge density; however, they all suffer from the same problem as the point matching test function: they do not capture the sharp peaks along the edges. Our metrics reflect this fact, as the mean absolute error for all three loss functions are larger than the baseline. We also see that the residual error distributions have larger variance and drift farther from a 0 mean. Our adaptive training scheme, therefore, makes training more accurate, while also reducing the necessary computations by gradually reducing the training dataset.

## 6 CONCLUSION, LIMITATIONS, AND FUTURE WORK

We propose a novel scientific machine learning technique leveraging a boundary element formulation of the Poisson equation to solve electrostatics problems. The method includes robust handling of the singularity introduced by the Green's function and an integration scheme that can easily be applied to quadrilateral meshes for solving arbitrary geometries. We study how the method scales and demonstrate that, unlike classical methods, the neural network basis is decoupled from the problem discretization, allowing the method to potentially scale to larger domains. This is a key benefit of PINN formulations, which leverage nonlinear mappings and gradient based optimization to circumvent the dense linear algebra solve that drive computation time. Ablation studies show that our choice of test function, singularity removal, and novel variational adaptive training scheme produce the strongest results. Neural electrostatics shows promise in solving general electrostatics problems, but it can still be optimized and improved to better rival classical methods. Training time of the technique is orders of magnitude longer than the BEM solver, as it uses highly optimized linear algebra solvers. There is also more to explore in test function design, such as using higher order polynomials. The collocation method could also be revisited with the residual-based adaptive distribution (RAD) (Wu et al., 2023) method, which improves upon RAR, but was out of scope for the current evaluations. Though Monte Carlo integration performed poorly in our tests, advanced technique such as control variates or importance sampling could be used to improve performance. We also emphasize that the proposed methods are insightful to a wide range of computational physics problems by adopting similar singularity removals or variational forms. Our method improves on previous learning approaches to 3D problems and develops a promising method for overcoming shortcomings of both classical and previous PINN methods.

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

## A    TRAINING

Training was done using RTX A5000 gpus on a compute cluster. Training averaged about 1 hour in length for each realization. Initial learning rates and batch sizes were determined using the Optuna (Akiba et al., 2019) tuner. Runs were trained for 1000 epochs using the PyTorch (Paszke et al., 2019) library.

## B    GREEN'S FUNCTION

Equation 1 is defined in 3D space and has no general closed form solution. Because the Poisson equation is linear, we can transform the PDE into an integral equation by convolving the impulse response of the Laplace operator with the forcing function, $-\frac{\rho_e}{\epsilon_0}$. This impulse response is known as the Green's function and it is found by solving

$$\nabla^2 G(\mathbf{r}, \mathbf{r}') = -\delta(\mathbf{r} - \mathbf{r}'). \tag{12}$$

For the Laplacian in a 3D open domain, $G(\mathbf{r}, \mathbf{r}') = \frac{1}{4\pi|\mathbf{r}-\mathbf{r}'|}$, with the boundary condition that the field must go to 0 at an infinite boundary. Applying the Green's function to Equation 1, we arrive at Equation 2. Sadly, this integral is often difficult and, in the general case, intractable. For one thing, $\rho_e(\mathbf{r})$ is unknown everywhere, while $\phi(\mathbf{r})$ is known only on a subset of the domain. If we are able to compute the charge density, then we can extend the voltage to anywhere in space using 2. The Green's function also contains a singularity when the field point, $\mathbf{r}$, approaches the source point, $\mathbf{r}'$. Luckily, this is a weak singularity (Colton & Kress, 2013) as its order is less than the dimensionality of the integral, so a solution can be found; however, it needs to be treated carefully as it can easily cause large errors in the solution.

## C    NEURAL STATICS BOUNDERY DERIVATION

Assume that we have a conducting surface, $\Gamma$, in free space. We are able to easily measure the voltage across the plate, however, we would like to characterize the electric field anywhere in space. Equation 1, at first glance, seems suitable for this problem; however, we do not know the charge density and only know the voltage along the conducting surface. We take inspiration from the classical boundary element method by using the Green's function and variational form simultaneously to restrict our problem to just that of the surface boundary. Equation 12 becomes a 2D surface integral given by

$$\phi(\mathbf{r}) = \iint_\Gamma \frac{\rho_e(\mathbf{r}')}{4\pi\epsilon_0|\mathbf{r} - \mathbf{r}'|} d\mathbf{r}'. \tag{13}$$

We chose to approximate the unknown function $\rho_e(\mathbf{r})$ along the plate by a feed forward neural network, $f_\theta(\mathbf{r})$, with ReLU activation functions. As $\phi(\mathbf{r})$ is known across the plate, we could randomly sample points and seek to equate the left and right hand sides through backpropagation. This formulation would create a flexible solver as only discrete samples from the surface are needed; however, as we show empirically in the results section, this results in a worse solution as the boundary condition is enforced only at these discrete points. Instead, we apply a rectangular testing function to enforce the boundary condition across the entire surface. This testing function is suitable for simple voltage functions and is used frequently in the classical method. It is easily represented by a quadrilateral mesh. The size of these quads are determined by the complexity of both the voltage function and the geometry.

## D    GAUSSIAN QUADRATURE AND CHANGE OF INTERVALS

In this section, we address two complications with evaluating Equation 9: the change of intervals and integrating a square. First, the Gauss-Legendre quadrature is defined on an interval from -1 to 1. To use this quadrature on other domains, the abscissas, $\alpha$, must be transformed using the change of interval formula. This simple formula is given by

$$\tilde{\alpha} = \frac{b-a}{2}\alpha + \frac{a+b}{2}. \tag{14}$$

Where, $a$ and $b$ are the left and right integral bounds, respectively.

Now we consider the integration over the square and why we use the notation $\tilde{\alpha}_{ji}$ in the main paper. To integrate a square in polar coordinates, it needs to be broken into 4 separate integral, where $\theta$ sweeps through 90 degrees at a time. The quadrants are defined along the diagonal cuts of the square (i.e., $[-\frac{\pi}{4}, \frac{\pi}{4}]$ and so on). Each integral then rotates this section by 90 degrees counterclockwise. The radius is then from 0 to $r\sec(\theta)$ (for left and right quadrants) or $r\csc(\theta)$ for the top and bottom. As the radius depends on $\theta$, the change of interval formula needs to be applied at every evaluated angle in the quadrature, hence $\tilde{\alpha}_{ji}$. By combining the contribution from each quadrant, we can compute the singularity and its surrounding domain in the near term integral.

The error associated with this approach is isolated to the Gauss-Legendre quadrature, which integrates the first $2N - 1$ terms in a Taylor expansion exactly. For hyperbolic tangent, this can be found in closed form, but ReLU makes this intractable; however, if the near term is contained within a single activation region of the ReLU network, the error will be very small as the quadrature needs to fit a straight line. The quadrature is not exact in this case due to the square boundary in a polar frame.

## E    Monte Carlo Far Term Integration

Monte Carlo integration is best suited for a near term that takes on a circular shape, as it can easily integrate the surface minus the disc around the singularity by resampling points that are in the near region. This technique suffers from the same problems stated in the singularity removal section: the error converges as the square root of the number of samples, and the network needs to learn the noise introduced by the estimate. These deficiencies, however, could be outweighed for more complicated models, but would most likely require importance sampling techniques to improve convergence. The Monte Carlo technique is also easy to implement and would make mesh free algorithms possible. In this form, the far integral becomes

$$\iint_{\Gamma-\epsilon} \frac{\rho_e(\mathbf{r}')}{4\pi\epsilon_0|\mathbf{r}-\mathbf{r}'|}d\mathbf{r}' = \mathbb{E}_{\mathbf{r}'\sim\sigma'}\frac{\rho_e(\mathbf{r}')}{4\pi\epsilon_0|\mathbf{r}-\mathbf{r}'|} \approx \frac{1}{N}\sum_{i=1}^{N}\frac{\rho_e(\mathbf{r}'_i)}{4\pi\epsilon_0|\mathbf{r}-\mathbf{r}'_i|}\cdot\frac{1}{\mathcal{P}(\mathbf{r_i})}. \tag{15}$$

$\sigma'$ is a distribution over $\Gamma - \epsilon$ and can be sampled easily using rejection sampling. $\mathcal{P}(\mathbf{r}'_i)$ is the probability of sampling that point in $\sigma'$. For uniform sampling, the probability is constant and the inverse of the surface area.

## F    Monte Carlo Singularity Exclusion

The Monte Carlo integration technique first recasts the integration in the form of an expectation, giving

$$\iint_{\Gamma} \frac{\rho_e(\mathbf{r}')}{4\pi\epsilon_0|\mathbf{r}-\mathbf{r}'|}d\mathbf{r}' = \mathbb{E}_{\mathbf{r}'\sim\sigma}\frac{\rho_e(\mathbf{r}')}{4\pi\epsilon_0|\mathbf{r}-\mathbf{r}'|}. \tag{16}$$

$\sigma$ is any distribution on $\Gamma$ and can be tailored to the specific surface geometry. In our work, we assume $\sigma$ to be a uniform distribution over the surface. It is important to remember that $\mathbf{r}$ is held fixed for each evaluation of the integral. Mathematically, the Monte Carlo form of the integral should be robust to the singularity, as the probability that the source and test point are the same during evaluation is zero. In practice, the expectation is still too singular to evaluate on a computer and some estimates will produce NaNs. The instability is mitigated by introducing a small exclusion radius, $\alpha \in \mathbb{R}^+$. If a sample is drawn within this disc, it is rejected and redrawn until it is sufficiently far from the singular point. This trick improves the stability of the estimation at the cost of introducing bias; thus, we seek to decrease the size of this exclusion region as much as possible while still

disallowing NaNs. The bias error is the difference between the exact integral, $I$, and the Monte Carlo approximation, $I'$. The difference is the integral of the disc of radius $\alpha$. By multiplying by the maximum value of the network on the disc, $f(\xi)$, the error is bounded from above, giving

$$|I - I'| \le 2\pi\alpha f_\theta(\xi). \tag{17}$$

Though easy to implement – there are many existing methods for sampling from a surface – the Monte Carlo technique has two downsides: its slow convergence and the stochasticity of its estimate. Monte Carlo techniques are known to decrease error as the square root of the number of samples. For example, to halve the error, we must compute quadruple the number of samples. Techniques such as control variates and importance sampling can improve the convergence, but they are more difficult to implement and must be tailored to specific geometries and problems. As each sample needs to be fed through the network and then backpropagated, the increased sampling requirement can quickly make the memory requirements prohibitively expensive. The stochasticity further complicates training by introducing noise into each evaluation of the integral. Though neural networks are robust to this type of noise, the randomness places more strain on the network as it has to learn both $\rho_e$ and the noise distribution.

## G   VARIATIONAL ADAPTIVE SAMPLING ALGORITHM

---

**Algorithm 1** Variational Adaptive Sampling

---
$epoch \leftarrow 1$
$model \leftarrow$ INITIALIZE( )
$data \leftarrow$ INITIALIZE( )
**while** $epoch \leq N_{stop}$ **do**
    $output, expected \leftarrow$ EVAL($model, data$)
    $loss \leftarrow$ COMPUTELOSS($output, expected$)
    BACKPROPAGATE( )
    **if** $epoch$ mod $N_d$ is 0 **then**
        $data \leftarrow$ REMOVE($data, |output - expected| \leq \delta$)   ▷ Remove data with smallest error
    **end if**
    **if** $epoch$ mod $N_r$ is 0 **then**
        $data \leftarrow$ INITIALIZE( )                        ▷ Switch back to full dataset
    **end if**
    $epoch \leftarrow epoch + 1$
**end while**

---

## H   HYPERPARAMETERS

| | Optimization | | | Adaptive Sampling | | | Singularity / | Test |
|---|---|---|---|---|---|---|---|---|
| Depth | Width | lr | Batch | $\delta$ | $N_r$ | $N_d$ | Integration | Samples |
| 2 | 64 | 0.0033 | 20 | 1e-6 | 15 | 90 | Transform + Grid | 100 |

Table 1: Hyperparameters used by the ablation baseline in the main paper. These are used as baseline parameters for ablations, which only seek to modify one of these values. If a column is not present in a future table, then all rows have the same value for that parameter, which can be found in this table. The singularity uses 2 by 2 quadrature points for each quarter of the subdivided square. The far terms are integrated in 10 by 10 squares with 3 by 3 samples points for Newton-Cotes quadrature.

| Optimization | | Adaptive Sampling | | | | Solver | | | |
|---|---|---|---|---|---|---|---|---|---|
| lr | Batch | $\delta$ | $N_r$ | $N_d$ | $N$ | Integration | Singularity | Test | $\epsilon$ |
| 5.879e-6 | 5 | 1e-6 | — | 677 | 20 | Monte Carlo | Transform | Collocation | 0.00103 |
| 0.0033 | 20 | 1e-6 | 15 | 90 | — | Grid | Transform | Point | $\sim 0.1$ |

Table 2: Hyperparameters used by the ablation study on different test functions. Most studies used MSE for the loss function and trained for 1000 epochs on a network with two hidden layers of width 64 each. The singularity removal transform uses a 2 by 2 quadrature on each of the four quadrants. The Monte Carlo integration uses 165 samples for the far integral. These samples are drawn randomly for each singular integration point. RAR was used to sample the points in the collocation case, and Optuna (Akiba et al., 2019) was used to find the hyperparameters, as the standard values did not converge well.

## I   COMPLEX EVALUATION

We include additional evaluations to show performance of more complex imposed voltages. In the first setting (Figure 6), we apply a sine and cosine to the x,y coordinate. The cosine has an amplitude of 1, an angular frequency of $\pi$ radians/s, and no phase offset. The sine has an amplitude of 0.5, an angular frequency of 7.2 radians/s, and a phase offset of 0.1 radians. There is also a DC component of 1 that raise the value at all points. These results demonstrate our work's ability to learn a basis function that is able to capture oscillations of multiple frequencies. Due to the spectral bias problem of low dimensional neural networks (Wang et al., 2020), we apply a Gaussian Fourier feature encoding (Tancik et al., 2020) to the input to aid in training. The dimension of this

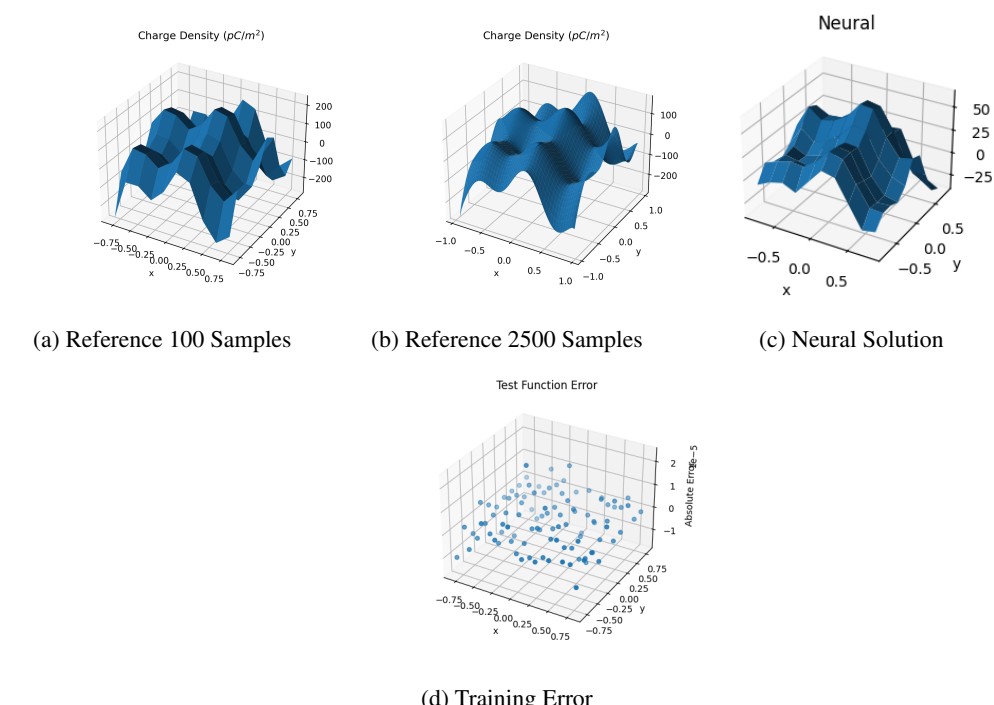

(a) Reference 100 Samples    (b) Reference 2500 Samples    (c) Neural Solution

(d) Training Error

Figure 6: We compare our neural solver to the BEM solver (Gibson, 2014) with a more complex imposed voltage containing multiple oscillatory frequencies. We see that our method is till able to capture the structure of the solution without increasing the number of testing functions.

embedding is 16 along each input coordinate, for a total of 32. The network size is the same as baseline, and Optuna (Akiba et al., 2019) is used to adjust training parameters, which minimized the error with: $lr$ of 0.001248, batch size of 34, $\delta$ of 0.0009634, $N_r$ of 19, and $N_d$ set to 72. We find that our method is able to match the shape of the solution, but there are some issues with the discontinuities around the edges, where the charge density does not increase enough.

In our second problem (Figure 7), we consider a ramp function for the voltage along the x-axis. This creates a non-differentiable discontinuity in the imposed voltage, which can cause difficulties in numerical solvers. Again, we use the Fourier feature encoding, and demonstrate that our method is able to handle these difficult problems. The network size is the same as baseline, and Optuna (Akiba et al., 2019) is used to adjust training parameters. The resulting values were: $lr$ of 0.001563, batch size of 39, $\delta$ of 0.003376, $N_r$ of 13, and $N_d$ of 49.

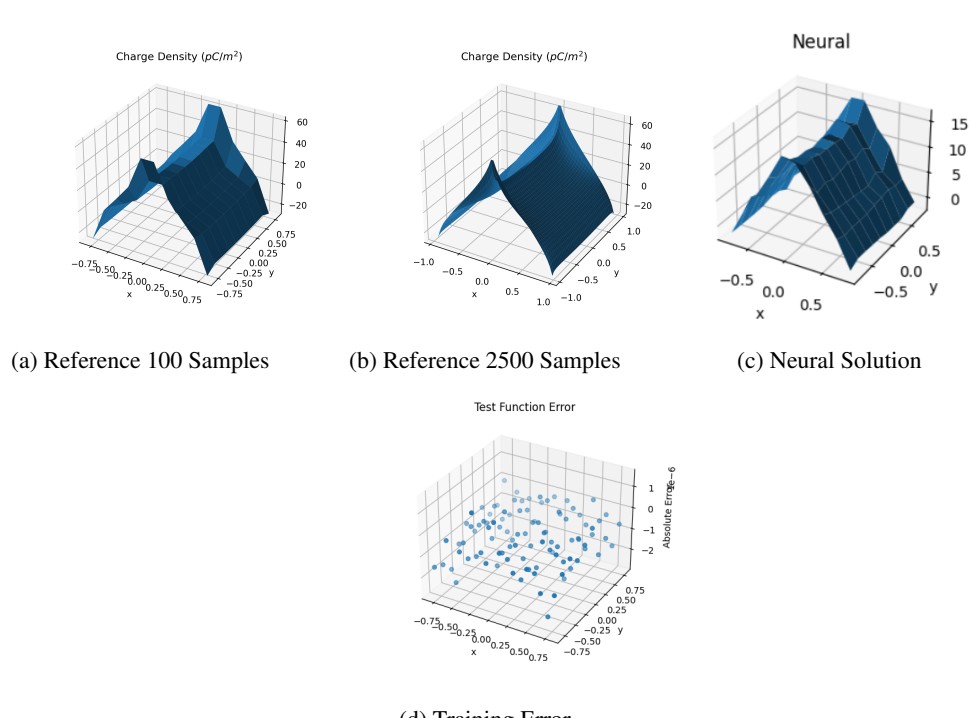

(a) Reference 100 Samples     (b) Reference 2500 Samples     (c) Neural Solution

(d) Training Error

Figure 7: We again compare the reference solution from the BEM solver (Gibson, 2014) to the results of our neural solver. We consider an imposed voltage in the shape of a triangle along the x-axis. The voltage is zero at x equal to -1 and 1, and 1 at x equal to 0. This creates a non-differentiable discontinuity in the domain, which can cause difficulties in numerical solvers. Our method is able to capture the shape of the solution even in this difficult case.

