# OpenReview forum: "Neural Electrostatics: A 3D Physics-Informed Boundary Element Poisson Equation Solver"
_ICLR.cc/2025/Conference — Submitted to ICLR 2025_

### Official Review · Reviewer_GWs9 · 2024-10-18

**Soundness:** 2
**Presentation:** 2
**Contribution:** 1
**Rating:** 3
**Confidence:** 3

**Summary:**

The paper aims at finding the density field in 3D electrostatic problems. This is done by modelling the density with a neural network, which is learnt in PINN fashion. Instead of solving the problem in its 3D original setting, the authors use known results from the literature that relate the solution in 3D to the solution at the 2D boundary, which is called boundary element method. This approach is known in the literature of PINNs but present some shortcomings. The contribution of the authors is to formulate the problem in spherical coordinates to avoid the singularity in the fundamental solution and to use a resampling scheme to improve the results.

**Strengths:**

- the idea of exploiting boundary elements method is indeed interesting both from a computational and conceptual point of view, even though it is not new in the literature
- the paper is easy to read

**Weaknesses:**

- the contribution of the paper is limited to the choice of spherical coordinates and of resampling test functions (which is however borrowed from PINNs trained with collocation points).
- I have the impression a classical numerical solver would anyway be preferable, especially in the simple (and only) setting used for evaluation.
- the paper is missing some important references to other PINN approaches that exploit the idea behind boundary element methods [1,2,3]. In particular, all papers study the Poisson equation, which is the focus of the present paper as well.

[1] Sun et al., BINN: A deep learning approach for computational mechanics problems based on boundary integral equations, Computer Methods in Applied Mechanics and Engineering, 2023

[2] Lin et al., BI-GreenNet: Learning Green’s Functions by Boundary Integral Network, Communications in Mathematics and Statistics, 2023

[3] Nagy-Huber et al, Physics-informed boundary integral networks (PIBI-Nets): A data-driven
approach for solving partial differential equations, Journal of Computational Science, 2024

**Questions:**

1. As argued in the paper, the proposed approach has the benefit of decoupling base functions from test functions with respect to classical numerical solvers. However, in the paper I did not find any strong reasons or discussion why PINNs should be used instead of the numerical solver. What is the main argument why the proposed approach should be preferable than the standard numerical solver? Is there any experiment where you show that for the same accuracy you require significantly less computational resources?
2. Did you actually mean that training time for the proposed method is orders of magnitude "longer" than classical numerical methods or did you mean "shorter"? line 535: "Training time of the technique is orders of magnitude longer than the BEM solver, as it uses highly optimized linear algebra solvers". This would make the BEM solver significantly more preferable than the proposed method.
3. The only experiment shown is very simple and by itself does not justify the use of the proposed approach. I believe that the paper would greatly benefit from further experiments and evaluation, possibly in less trivial settings
4. Furthermore, the comparison with PINNs would require some more in-depth analysis. I would expect that when many collocation points are used (as many as they can fit in the GPU) a somewhat reasonable solution is learnt. Do you observe this? if so, it would be interesting to see a comparison as a function of the collocation points used. Recent work RAD [4] improves on RAR and would be a relevant comparison for the present method.
5. I am a bit confused about the singularity argument in Equation (7). If I understand correctly, with spherical coordinates the singularity in the denominator is absorbed in the singularity of the coordinate system, so one can compute the integral without worrying about it. However, doesn't this hold only for point sources?
6. I also have a broader question concerning the boundary element method for Equation (1). Isn't the dimensionality reduction from 3D to 2D only possible for point sources? or anyways for sources that allow to compute the source integral easily?

I found the following typos in the paper:
- 88 "decision decisions"
- 182 "v_i, ..., v_N" --> should be "v_1, ..., v_N"
- 202 "Singularity in 12"  --> I guess you meant 2 and I would add "Eqaution" as you did throughout the paper
- 203 "simultaneous" --> "simultaneously"
- 338 I would suggest to use another notation since epsilon was already used in Equation 8
- 431 "test function" --> "test functions"
- 445 Figure 4 does not have a caption saying that it is Figure 4

[4] Wu et al., A comprehensive study of non-adaptive and residual-based adaptive sampling for physics-informed neural networks, Computer Methods in Applied Mechanics and Engineering, 2022.

---

> ### Author Response · Authors · 2024-11-25
> **Response to Weaknesses**
>
> Thank you for your detailed feedback and for highlighting both the strengths and areas for improvement in our manuscript. We appreciate the opportunity to respond and have carefully considered your concerns and questions. We outline the steps we are taking to improve the work in an upcoming draft.
>
> **Weaknesses**
>
> 1. The paper is limited to spherical coordinates and resampling test functions
>
> We ask that the reviewer clarify this concern more if we do not address it adequately. While we use a local coordinate transform to remove the singularity, this choice does not inherently limit the applicability of our approach. We simply remove the singularity locally, which greatly improves the stability of training, as shown by our ablation study. The terms, away from the singularity, are integrated as expected in a Euclidean frame. The variational sampling technique is also important for stability. Unlike previous work (Lu et al., 2019; Wu et al., 2022), our technique does not increase training time because it only decreases the training dataset instead of adding new points.
>
> 2. Classical solver is preferred
>
> We agree that classical solvers have been optimized for many problems; however, like past PINN work, we use the classical solver (Figure 1) as a baseline to validate our answer. PINNs in general struggle to beat classical solvers, but classical solvers also have decades of optimization. Our method provides improvements over previous PINN techniques (Section 5.3), on which the field can build to make the methods more competitive.
>
> 3. Missing reference to important works
>
> We apologize for the oversight in omitting these important references. We will include these papers in our updated "Related Work" and "Results" section. These works provide valuable contributions to the field, but there are key differences compared to our approach. BINN focuses on 2D domains and does not easily generalize to arbitrary 3D shapes, employs a different singularity removal strategy, and lacks a variational form, which we found to be critical for stability in our ablation studies. Bi-GreenNet fits more within the operator learning literature, aiming to learn a class of functions rather than solve a single instance of a PDE. BiNet, by the same authors as Bi-GreenNet, is a closer comparison to our work. BiNet lacks a variational formulation, evaluates only on a 2D domain, and does not directly address the singularity removal. Lastly, PIBI-Nets use Monte Carlo evaluation, which we found to unstable in our ablation study and focuses on both $\phi$ and $\rho$ being unknown. All three of these works form their boundary integral using potential theory, whereas we focus on a form that is of more interest to the computational electromagnetics community. These works are all covered by our ablation studies, as they use the collocation method. They also only evaluate on a 2D domain, while we can handle an arbitrary 3D quadrilateral mesh.

---

> ### Author Response · Authors · 2024-11-25
> **Response to Questions**
>
> **Questions**
>
> 1. Why should PINNs be used instead of numerical solvers?
>
> While classical numerical solvers have benefited from decades of optimization, PINNs offer distinct advantages that make them valuable for specific problem classes. PINNs compress the solution space into a small neural network, making them particularly suitable for scenarios requiring repeated evaluations or parameter sweeps. By using a boundary form, our approach eliminates the need to mesh the entire domain and enables field evaluations at arbitrary points in space, which is a significant advantage over traditional solvers. Furthermore, our single loss term formulation avoids the instability commonly associated with balancing multiple terms in traditional PINNs, resulting in a more stable and efficient training process. Through this work, we aim to advance the development of PINNs to enhance their competitiveness and utility in solving challenging problems.
>
> 2. Is training longer than the classical solve? Is the BEM always preferable?
>
> We acknowledge that training time for PINNs can be longer compared to classical solvers, but our method introduces improvements that make PINNs more competitive. Unlike traditional BEM, which requires assembling and solving a full $N\times N$ matrix, our PINN-based approach avoids this step, offering potential scaling benefits, especially for large problems. This reduction in computational requirements aligns with the broader goal of optimizing PINNs for real-world applications where classical methods may become computationally prohibitive. We find that our training tim is on the order of 50 minutes for most problems, but we were focused mainly on correctness as opposed to optimization of training time, which is left to future work.
>
> 3. Can better evaluations be provided?
>
> Yes, we will address this issue in the revised manuscript by including additional evaluations that consider more complex voltage functions with multiple frequencies and non-differentiable discontinuities. These expanded evaluations will demonstrate the robustness and general applicability of our method to a wider range of scenarios, beyond the simple settings presented in the current draft.
>
> 4. Can comparisons to RAD or a large number of collocation points be provided?
>
> Adding more collocation points could yield a more favorable comparison but would place traditional collocation methods at a disadvantage relative to our approach, which achieves strong performance with far fewer test elements. We will look at adding RAD to our ablation study in the upcoming draft. Unlike RAD, which increases the training dataset size and slows down training, our variational adaptive scheme reduces the size of the training dataset, resulting in faster training and improved accuracy. This efficiency makes our method a compelling alternative for large-scale problems.
>
> 5. Does the singularity removal only hold for point sources?
>
> Yes, the form of our singularity removal is only valid for point discontinuities as found in the Green's function in the electrostatics problem. This singularity removal is specific to the problem formulation but has broad applicability in computational electromagnetics, where $\phi$ (the boundary potential) is known, and $\rho$ (the charge density) is unknown. This targeted singularity removal significantly enhances the stability of the solution process, as demonstrated in our ablation studies, compared to simple Monte Carlo integration.
>
> 6. Isn't the dimensionality reduction from 3D to 2D only possible for point sources?
>
> In the context of electrostatics, the boundary can be viewed as a surface composed of infinitesimal point sources, each radiating an electric field according to the Green's function and subject to zero boundary conditions at infinity. This dimensionality reduction is a natural consequence of the boundary integral formulation and is critical for many practical applications like capacitive touch screen design (Zhaou et al., 2020) or general electrostatic simulation (Gibson, 2014)  in computational electromagnetics. Our approach aligns with these applications and leverages this formulation to provide efficient and accurate solutions, as discussed in the introduction.
>
> We appreciate your thoughtful feedback and have worked to address all concerns and improve the clarity and importance of our manuscript. These updates, alongside additional evaluations and comparisons, will be reflected in the upcoming draft. Thank you again for the time and effort spent reviewing our work.

---

> ### Comment · Reviewer_GWs9 · 2024-11-27
>
> I would like to thank the authors for their answers to my concerns. I will follow-up below on a few points mentioned.
>
>
> ### Weaknesses
> > 1. The paper is limited to spherical coordinates and resampling test functions
>
> I am afraid we had a misunderstanding there. I meant that the contribution of the paper (in terms of novelty) are limited to the use of spherical coordinates and to the use of resampling functions (which is a standard technique for collocation points). I still believe that these two are the main contributions of the work.
>
> > 2. Classical solver is preferred
>
> I agree that classical solver have benefited from years of research. However, PINNs are still very valuable in situations where numerical solvers struggle (which is not the case in the present work). For instance, numerical solvers struggle to integrate information about data, which in contrast can be easily integrated in PINNs. Since the proposed method is not explicitly compared to any other work it is hard to see the improvement and one still wonders why the numerical solver should not be used in practice anyways (since it is apparently even much faster). Also, other works like BINN or PIBI-Nets either show extremely accurate reconstruction or have a setting where numerical solvers cannot be used directly.
>
>  > 3. Missing reference to important works
>
> I thank the authors for including the mentioned literature in the upcoming revised pdf (which I am looking forward to reading).

---

> > ### Author Response · Authors · 2024-11-28
> >
> > 1. The paper is limited to spherical coordinates and resampling test functions.
> >
> > Thank you for clarifying this point, and we apologize for the confusion. We agree that the coordinate transform singularity removal and variational adaptive sampling are key contributions of our work, but we would also like to emphasize the boundary integral adapted to the electrostatics problem instead of derived using potential theory as well as the extensive comparisons between different loss functions, singularity removal, and integration schemes. Past work has not been as readily applicable to the computational electromagnetics community, nor has it demonstrated the difficulty of using other methodologies to train their networks. Also, though the variational adaptive sampling draws inspiration from past techniques, it is a key enabler of larger problems as it reduces the sampling requirements during training while also improving error in regions that need it the most.
> >
> > 2. Classical solver is preferred.
> >
> > We believe that our work is most comparable to BINN; however, we solve a different boundary integral and use a variational form. BINN compares their results to an FEM solver, but does not give their evaluation times.  BINN does find that they can use fewer sample points for a better solution, which is also what we found in our scaling experiments. We go a step farther and extend the results to 3D and use a variational form. Though we are not immediately faster, than classical methods, there is hope that a variational boundary element solver like ours could allow for easier extension to larger, complex domains than classical solvers are able to handle today.
> >
> > 3. Missing reference to important works.
> >
> > We highlight these comparisons in the updated results section; however, no additional ablation studies were run. The other methods are covered by the collocation scheme with Monte Carlo integration of the far terms. The only comparison we do not have is to the singularity extraction technique used in BINN, due to time constraints.

---

> ### Comment · Reviewer_GWs9 · 2024-11-27
>
> ### Questions
>
> > 2. "We acknowledge that training time for PINNs can be longer compared to classical solvers"
>
> I would suggest the authors to explicitly include in the paper a discussion about computational costs of the proposed approach and a comparison with numerical solvers. Unfortunately, if the proposed method is not as accurate as numerical solvers and still takes significantly longer than numerical solvers, it is hard to think that it could be advantageous in practice. This is especially true if, compared to numerical solver, the proposed approach takes "orders of magnitude longer".
>
> However, in some settings (e.g. when data is also available) you might be able to argue that your method is preferable to numerical solvers. Also, advancement with respect to other competitive PINN approaches should be shown explicitly in the paper and in a way that highlights limitations and advantages of the proposed approach.
>
> > 3. Can better evaluations be provided?
>
> I thank the authors for working on additional experiments, which should not be given for granted.
>
> > 4. "Adding more collocation points could yield a more favorable comparison but would place traditional collocation methods at a disadvantage relative to our approach"
>
> Then I would suggest to show the error with respect to the ground truth for fixed computational resources. If it is true that "variational adaptive scheme reduces the size of the training dataset, resulting in faster training and improved accuracy" then it should be easy to see by setting a fixed number of computational resources and use for RAD as many collocation points as to reach that value. Or do you in general achieve improved accuracy?
>
> > 5./6. Isn't the dimensionality reduction from 3D to 2D only possible for point sources?
>
> I still don't see how the dimensionality reduction is possible only for point sources but can be extended to any surface as long as it is considered as a set of point sources. Is the surface treated as a (finite) collection of points sources or is the argument made in the limit of infinitesimal point sources? in the first case I agree it would work (but then this should be mentioned explicitly in the paper) while in the second case I still don't get how it would work.

---

> > ### Author Response · Authors · 2024-11-28
> >
> > 2. Comparison to numerical solvers.
> >
> > We thank the reviewer for helping to improve the comparisons in our work. As shown by the scaling study (Section 5.2) and the ground truth mean absolute error (Figure 4a, blue bar). We see that a given a small number of test functions, our neural solution matches closer to the ground truth solution than it does to the reference BEM solver with the same number of test functions as training. As such, we find that the neural solver with variational adaptive sampling and coordinate transform singularity removal compares favorably in terms of accuracy, while using fewer test functions. To emphasize this important aspect of our work, we have added additional explanations to the scaling study (Section 5.2) and the conclusion (Section 6).
> >
> > The end of section 5.2 now reads:
> > "Future work could leverage this decoupling to solve larger problems with far fewer basis and test elements than is traditionally needed."
> >
> > Section 6 includes a sentence that states:
> > "This is a key benefit of PINN formulations, which leverage nonlinear mappings and gradient based optimization to circumvent the dense linear algebra solve that drive computation time."
> >
> > 3. Additional evaluations
> >
> > New results can be found in Appendix I, due to space constraints in the original paper. We will clean up the plots in the final draft.
> >
> > 4. Comparison against collocation point method.
> >
> > We thank the reviewer for highlighting this critical aspect of our contributions and working to improve its comparison to previous literature. This is what we sought to show in the test function ablation study (Section 5.3.2). RAR was used and Optuna was allowed to optimize the loss over 20 training trials to give the collocation method a reasonable advantage. Even so, we found that the collocation method greatly under performed the variational method. The variational scheme also provides an easy means to perform singularity removal, as it provides an easy-to-use grid for the quadrature. The free form nature of the collocation method almost necessitates the use of Monte Carlo integration of the far term. This limits reuse of network evaluations, and as we showed in Section 5.3.3 greatly under performs the Newton-Cotes integration of the far term. Because of these reasons, we believe that the variational training is important for achieving greater accuracy with less resources. There is also some intuition from the BEM literature that the variational method is imposing the boundary voltage over a large area, as opposed to just single points. We also would like to note that we are unable to provide RAD evaluations by the revision deadline; however, as RAD is still a collocation method, we expect similar results. We will mention this lack of comparison as one of the limitations (Section 6). We have also expanded the discussion in the test function ablation study (Section 5.3.2) to better emphasize this comparison.
> >
> > Section 6 update:
> > "There is also more to explore in test function design, such as using higher
> > order polynomials. The collocation method could also be revisited with the residual-based adaptive distribution (RAD) method, which improves upon RAR, but was out of scope for the current evaluations."
> >
> > Section 5.3.2 update:
> > "Compared to our baseline, for a similar number of data points, we find that the variational scheme is able to reduce the PDE error more effectively. We also note that RAR continually adds points to the dataset, which can drastically impact training, as these additional points require a singularity removal and integrating across the boundary. Our variational adaptive sampling method instead reduces the number of evaluations while also decreasing error."
> >
> > 6. How is the dimensionality reduction possible?
> >
> > Yes, the surface can be viewed in the limit of infinitesimal point sources. This is common in computational electromagnetics (Gibson, 2014). Each point on the surface radiates in all directions based on the free space (infinite domain) Green's function. The sum of all these point source contributions represents the field anywhere in space. For a more concrete problem, consider a flat metal plate with an imposed voltage. The plate is surrounded by air, which is commonly modeled as free space, which does not hold a charge. The voltage is only imposed on the plate, and the charge density is only non-zero on this plate as well. This is a common setup for looking at capacitor design and is a common baseline for method of moments problems (Gibson, 2014).
> >
> > We thank the reviewer for their time and appreciate the constructive feedback.

---

> ### Comment · Reviewer_GWs9 · 2024-12-01
>
> ## Weaknesses
>
> 2. "Though we are not immediately faster" sounds very different from "orders of magnitude longer". Anyways, I think that in order for the proposed method to be useful in practice it must be either more accurate or faster. Else, it might be that either the particular application setting or the method are not suitable, in my opinion. If your point is that your proposed method might be beneficial in "larger, complex domains", then this should be reflected in the experiments as well. At least showing some trend as the dimension/complexity increases.
>
> 3. In PIBI-Net it seems that the Monte Carlo integration works perfectly, so it might be that it depends on how points are sampled (particularly those close to the boundary) or by the specific setting under study.

---

> > ### Comment · Reviewer_GWs9 · 2024-12-01
> >
> > ## Questions
> >
> > 2. To me what the comparison with the BEM solver shows is that with not enough (expressive) test functions the solution is not properly learnt. However, this doesn't mean that the BEM solver cannot be used to find a solution in significantly less time and that is more accurate than the one of the proposed method (granted that enough test functions are used - but this doesn't constitute a computational bottleneck)
> >
> > 3. I thanks the authors for the additional evaluation and for putting extra effort during this rebuttal. I think the new experiments are similar in nature to the ones already present in the main text. I believe that what would need to be shown is that the proposed method is more accurate (at fixed training time) or faster (at fixed - and sufficiently high - accuracy) than the BEM solver. If the numerical solver continues to be more accurate and comparably fast (or in this case significantly faster), I don't think a PINN alternative should be used.
> >
> > 4. I thank the authors for expanding on the comparison with RAD. Also in this case the argument should be made more quantitative and show that at fixed computational resources the proposed method achieves better (and sufficiently good - not just better) accuracy with respect to RAD. By itself I don't think that "similar number of data points" is a fair comparison, since the two methods work differently - but I might be wrong on this point. From a technical perspective is clear however that avoiding integrating over the boundaries can be beneficial (even though it also depends on how the samples are taken, see previous comment about PIBI-Net)
> >
> > In the updated version of the manuscript please take care to update the figures such that the labels and ticks are readable in A4 format. At the moment, almost all figures are hard to read.

---

### Official Review · Reviewer_iy3h · 2024-10-30

**Soundness:** 2
**Presentation:** 3
**Contribution:** 2
**Rating:** 5
**Confidence:** 2

**Summary:**

The paper proposed a novel electrostatic solver by combining an adapted physics-informed neural network (PINN), and a 3D boundary element method (BEM). With special techniques, the solver can simplify the original 3D problems into 2D problems, and solve them more effectively. The results are demonstrated on an experiment of a flat plate with constant charge, and with different test function configurations.

**Strengths:**

- The paper is easy to follow, well conveyed, and I appreciate that.
- The proposed 3D boundary element solver is able to combine domain and boundary losses into a single loss.
- By transforming the integral equations into polar coordinates, the solver can remove singularity.
- A variational sampling techniques is introduced to improve accuracy of the solver.

**Weaknesses:**

- I did not see any demonstration of the neural network architecture. A diagram would help. Implementation details such as training data size are also missing.
- I expect more contents in the result section. More experiments are needed to evaluate the neural network model. For example, different geometry domains, different scale, etc.

**Questions:**

- Is the neural solver able to generalize to different domains, ie, different shape or size?
- To judge the practicality of the solver, do you have inference time recorded? Is it faster than the classical method?

---

> ### Author Response · Authors · 2024-11-25
> **Response to Weaknesses and Questions**
>
> Thank you for your thorough and constructive feedback. We appreciate the time spent to review our work and help strengthen our contributions. Below, we address your concerns and questions. We also include key additions that we will make to the revised draft.
>
> **Weaknesses**
> 1. No architecture diagram
>
> We apologize for the omission of a diagram illustrating the neural network architecture. We will include a clear diagram in a revised draft, along with extended tables detailing all hyperparameters in the appendix.
>
> 2. Lack of evaluations
>
> Our ablation studies on simple domains demonstrate the robustness of our approach compared to previous PINN methods, even in simple domains. We will include results with more complex imposed voltages. One example imposes a voltage of a series of sines and cosines to show the network's ability to capture multifrequency details. The second example uses a ramp function to show results with a non-differentiable imposed voltage. The derivation is also general and supports any 3D geometry represented by a quadrilateral mesh, but our evaluations consider a rectangular domain to ease implementation of mesh free methods. Additionally, we would like to emphasize that most prior PINN methods restrict their evaluations to 2D domains, whereas we specifically tackle general 3D surfaces.
>
> **Questions**
> 1. How does the method extend to different domains (both size and shape)?
>
> Yes, our derivation applied to general 3D quadrilateral meshes. We restrict our domain to a rectangle to ease initial implementation; however, the sharp boundary (Figure 3) already shows how the method reacts to non-differentiable discontinuities in the domain. We will include additional evaluations on more complex forcing functions in the upcoming draft.
>
> 2. Do you have inference time recorded? Is it faster than the classical method?
>
> We will include inference times for integrating the network across the surface, which should be on the order of about a second. To provide some context:
> - Evaluations require integrating across the entire domain, which requires many evaluations of the network. This is shared by all boundary element formulations.
> - The neural evaluations are trivially parallelizable, as batch process is supported by all modern machine learning frameworks.
> - Classic solvers have benefited from 60 years of optimization, whereas PINN are still trying to find their footing.
> - Using the neural solution and boundary element form allows for the problem out to infinity to be heavily compressed.
> - Our work provides an improvement over existing PINN methods as the field seeks to improve over the existing classical methods.
>
> We appreciate your insightful feedback and believe these updates will strengthen the manuscript. Thank you again for your valuable suggestions, and we would be happy to address any other questions or concerns. We will send a followup message once the new draft is ready.

---

> > ### Comment · Reviewer_iy3h · 2024-11-26
> >
> > Thank you for your reply. However, I decided to maintain my score.

---

### Official Review · Reviewer_vXUM · 2024-11-03

**Soundness:** 2
**Presentation:** 3
**Contribution:** 2
**Rating:** 3
**Confidence:** 4

**Summary:**

This work applies PINN in a boundary element form to solve electrostatics problems. The singularity issue is handled by locally transforming the integral into polar coordinates. A variational adaptive sampling method is also introduced to improve the solving efficiency. Experimental results show the proposed method can obtain results similar to a traditional BEM solver while using 25 times fewer elements.

**Strengths:**

- The idea of this work is reasonable in principle.
- The writing is overall clear and detailed.

**Weaknesses:**

- This paper limits itself to solving electrostatics problems. On the other hand, it can be easily extended to a wider range of scenarios in the computational mathematics sense, so that it will be more appealing to a much wider audience of ICLR.
- Combining PINN and the boundary integral equation has been investigated by a few previous works, e.g., [1]. However, there are no analytical or experimental comparisons provided in this paper.
- The experiment is overall too simple and not very convincing. The provided test case is just 1 specific example, which is not very difficult in scale and in smoothness. Also, there lacks comparison to previous works, for example PINN variants designed for BEM.

[1] Lin, Guochang, et al. "BINet: Learning to solve partial differential equations with boundary integral networks." arXiv preprint arXiv:2110.00352 (2021).

**Questions:**

/

---

> ### Author Response · Authors · 2024-11-25
> **Response to Weaknesses and Questions**
>
> Thank you for your thoughtful feedback and for taking the time to evaluate our work. We have carefully considered your comments and addressed the concerns and questions below. We outline our plans to address these concerns in a revised draft.
>
> **Weaknesses**
> 1. Limited scope to the electrostatics problem.
>
> We highlight the connection to electrostatics because we demonstrate the use of a common boundary integral from the computational electromagnetics (CEM) literature (Gibson, 2014). This approach differs from the potential theory based integral often found in other boundary element based PINN methods (Lin et al., 2021; Sun et al., 2023; Nagy-Huber and Roth, 2024). One key difference is that the forcing, $\rho$, is the unknown quantity as only $\phi$ is easily measured. As suggested, however, the analysis and techniques for singularity removal (Section 4.2) and variational adaptive sampling (Section 4.4) are not limited to electrostatics. Our hope is that our paper (esp. the title) would motivate our method for general Poisson equation solvers. We will seek to highlight these extensions as part of the future work.
>
> 2. No comparisons to previous works on boundary element based PINNs.
>
> We apologize for missing direct comparisons to other works like BINet (Lin et al., 2021), and will include it, along with several others, in the "Related Work" section (Sun et al., 2023; Nagy-Huber and Roth, 2024). These works, however, are covered already in the ablation studies (Section 5.3), which show that our proposed boundary method with a variational loss and coordinate transform singularity removal outperforms previous methods. We will make this comparison more explicit in our results section. For BINet specifically, the main differences are:
> - BINet does not use a variational form, which we found to be crucial for stability. BINet uses the collocation method without any adaptive sampling, which can be found in Figure 4a of section 5.
> - BINet uses the potential theory form of the boundary integral, which is not as directly applicable to the electromagnetics community.
> - BINet only performs evaluation on a 2D domain, where we provide an extension to general 3D quadrilateral surface meshes.
> - BINet does directly address the singularity introduced by the Green's function. We show and compare multiple techniques for handling this common difficulty. Without care, the singularity can cause instability in training.
>
> 3. Evaluations are too simple.
>
> Our ablations were completed on a simple domain to demonstrate that many commonly used techniques struggle even in this straightforward setting. Past work also restricted themselves to simple 2D domains for evaluation, whereas we learn a solution on an arbitrary quadrilateral 3D mesh, already a standard requirement for classic physics solvers. We also recognize that more evaluations would strengthen our results, and have included additional evaluations on complex imposed voltages to show the generality of our technique to functions with multiple frequencies and non-differentiable discontinuities.
>
> We appreciate your constructive feedback and believe that these improvements to our results and inclusion of additional literature will enhance the relevance and impact of our work. Thank you again for your time. We would be happy to address any more questions or concerns and will send another message once the new draft is ready.

---

> ### Comment · Reviewer_vXUM · 2024-11-30
>
> Thank you very much for your hard work on addressing the reviewers' comments. I went through all the discussions. There are a few concerns shared by all reviewers, e.g., lack of comparison to previous works, the complexity and abundance of the experimental settings, the clarification of the details, etc. With all factors considered, I tend to maintain my score.

---

### Official Review · Reviewer_8ZGv · 2024-11-04

**Soundness:** 2
**Presentation:** 2
**Contribution:** 2
**Rating:** 5
**Confidence:** 3

**Summary:**

The authors present a neural solver for eletrostatics problems (Poisson's equation) where the loss function is inspired by the boundary element methods. Using this formulation, both the domain and the boundary integrals feature in the residual together, and therefore, no special weighing of domain and boundary terms is needed. A singularity appears in the boundary integrals, and the authors use a change of coordinate system for the boundary integral to handle this. The authors present their results on one particular example, and provide ablation studies on various test functions, loss functions and integration techniques.

**Strengths:**

The paper is well written.

I like how the $ i^{th} $ residual (Eq 5) is a linear combination of a domain integral ($ \textlangle v, \phi \textrangle $), and another term that contains a boundary integral. This means, the residual for each test function has a naturally formulated contribution from the domain and the boundary. Therefore, no extra weighing is needed between the boundary and the domain terms.

In scaling study, it is interesting to see that the method is largely independent of the size of the network (among the sizes tested).

**Weaknesses:**

A lot of the details are unclear. How does the mapping look like? Does the network take in the coordinate of a point as input, and return both $ \phi $ and $ \rho $? The mapping detail, and the architecture should be made more precise. A flow-chart / picture might help.

Only one simple problem is considered in the results section.

Minor:

A lot of abbreviations used without explanation.

Sec 2 header: "work"

line 134: vein instead of vain

**Questions:**

Please specify ''rectangular test function'' precisely.

Fig 2: why are the errors for 225 test functions higher than both 25 and 100 test functions?

Please state the boundary conditions for the example considered in Sec 5.

The authors use the so-called "fundamental solution'' as Green's function in this paper. The fundamental solution pertains to the PDE solved on $ R^n $ with no boundary condition. But the considered example is a finite domain with some boundary conditions (though the boundary conditions are never stated). Does the Green's function remain the same as the fundamental solution in this case? Could the authors please explain this in detail?

---

> ### Author Response · Authors · 2024-11-25
> **Response to Weaknesses and Questions**
>
> Thank you for your thoughtful review and constructive feedback. We appreciate the time and effort taken to review our work, and have carefully considered your comments to improve the manuscript's impact. Below, we address the weaknesses and questions, and outline changes that will be present in an upcoming draft.
>
> **Weaknesses**
> 1. Details about architecture and training are unclear.
>
> The network takes in either uv mapped 2D or Euclidean 3D coordinates and outputs just the scalar value $\rho$. The imposed boundary voltage is given by $\phi$. This problem setup is common in electromagnetics (Gibson, 2014), where the forcing is the unknown parameter. To improve understanding, we will include an architecture diagram with network inputs, network outputs, and training loss computation.
>
> 2.  Only a simple problem is considered
>
> A simple problem was used to demonstrate that traditional PINN methods (Raissi et al., 2017; Lu et al., 2019) using collocation points and adaptive sampling struggle to obtain the solution, even in basic scenarios; however, we recognize the importance of broader evaluation. We will include results with more complex imposed voltages. One example imposes a voltage of a series of sines and cosines to show the network's ability to capture multifrequency details. The second example uses a ramp function to show results with a non-differentiable imposed voltage. We also note that our derivation is suited for arbitrary 3D meshes and many other works in the literature (Lin, et al. 2021; Raissi et al., 2017; Lu et al., 2019) have restricted their evaluations to simple boundaries in 2D domains.
>
> 3. Abbreviations Used without Explanation
>
> We apologize for the confusion and will ensure that all abbreviations are clearly defined on first use.
>
> 4. Spelling errors
>
> Thank you for spotting these errors. They will be corrected in the revision.
>
> **Questions**
> 1. Specify the rectangular test function precisely.
>
> The rectangular test function, commonly use in finite element methods (Gibson, 2014), is defined as having a single value with support over a compact element. In our case, this support is a single quadrilateral in the mesh. We will include a precise description in the paper to improve presentation.
>
> 2. Why are the errors for 225 test function higher than both 25 and 100 test functions?
>
> The errors for 225 test functions are higher because there are more evaluations near the discontinuities along the domain edges. Capturing these discontinuities is inherently challenging for any computational method and can increase the error metric. Computational methods circumvent this by integrating farther from the discontinuity. We will add a note on this observation in the revised draft.
>
> 3. Please state the boundary conditions for the example considered in Section 5.
>
> Our analysis assumes the common exterior electrostatics problem of an infinite domain, where the field decays to 0 along a boundary at infinity. This setup is a significant advantage of boundary element methods, as the solution can be found anywhere around the surface. We will explicitly state these assumptions in the revised draft.
>
> 5. What is the Green's function used in the example, and how does it differ from the fundamental solution?
>
> As noted above, we focus on the infinite domain form of the problem, leveraging the benefits of the boundary element method to find solutions anywhere in space. In this setting, the Green's function is the same as the fundamental solution for the Laplace operator. The method can be adapted to interior domains with the appropriate Green's function. Our results show the charge densities plotted on a flat plate in the domain (Figure 3). By integrating these charge densities, the electric field can be found anywhere. We will discuss these adaptations and limitations in the revised future work section.
>
> Thank you again for the detailed feedback. These revisions will strengthen our manuscript and clarify the contributions of our work. We would be happy to address any more questions or concerns that you have and will send another message once the new draft is ready.

---

> > ### Comment · Reviewer_8ZGv · 2024-11-26
> >
> > Thank you for answering my questions. My impression of the manuscript remains unchanged, so I will maintain my score.

---

### Meta-Review · Area_Chair_zncw · 2024-12-20

**Metareview:**

The paper proposes a neural approach for solving electrostatic problems (essentially modeled by the Poisson equation) which combines the well-known physics-informed neural network (PINN) framework with a 3D boundary element method. The idea is intuitive, the paper is mostly well-written and the authors support the idea with a few experiments.

A few common concerns were raised about the current state of the paper:
* several missing technical details
* inadequate discussions and comparisons to closely related previous work that also combine PINNs with BEMs
* the experiments. To me this was the biggest shortcoming: the authors only considered a simple toy physical system for a simple geometry. I would have expected a much more diverse collection of experimental evidence.

Based on the above reasons, I recommend a reject, while advising the authors to keep the above feedback in mind for future revisions.

**Additional Comments On Reviewer Discussion:**

The reviewers mostly echoed the weaknesses listed above. There was some back and forth between the authors and the reviewers, but opinions remained unchanged.

---

### Decision · Program_Chairs · 2025-01-22

Reject